:ʘ: PLOS | ONE

# Phylogenetic analysis reveals an ancient gene duplication as the origin of the MdtABC efflux pump

Kamil Górecki [1], Megan M. McEvoy [1,2,3]*

**1** Institute for Society & Genetics, University of California, Los Angeles, California, United States of America, **2** Department of Microbiology, Immunology & Molecular Genetics, University of California, Los Angeles, California, United States of America, **3** Molecular Biology Institute, University of California, Los Angeles, California, United States of America

* mcevoymm@ucla.edu

**Data Availability Statement:** All relevant data are within the manuscript and its Supporting Information files.

**Funding:** The author(s) received no specific funding for this work.

## Abstract

The efflux pumps from the Resistance-Nodulation-Division family, RND, are main contributors to intrinsic antibiotic resistance in Gram-negative bacteria. Among this family, the MdtABC pump is unusual by having two inner membrane components. The two components, MdtB and MdtC are homologs, therefore it is evident that the two components arose by gene duplication. In this paper, we describe the results obtained from a phylogenetic analysis of the MdtBC pumps in the context of other RNDs. We show that the individual inner membrane components (MdtB and MdtC) are conserved throughout the Proteobacterial species and that their existence is a result of a single gene duplication. We argue that this gene duplication was an ancient event which occurred before the split of Proteobacteria into Alpha-, Beta- and Gamma- classes. Moreover, we find that the MdtABC pumps and the MexMN pump from *Pseudomonas aeruginosa* share a close common ancestor, suggesting the MexMN pump arose by another gene duplication event of the original Mdt ancestor. Taken together, these results shed light on the evolution of the RND efflux pumps and demonstrate the ancient origin of the Mdt pumps and suggest that the core bacterial efflux pump repertoires have been generally stable throughout the course of evolution.

## Introduction

The resistance-nodulation-division efflux pumps (RNDs) comprise a large family of proteins, widely distributed among bacterial species [1,2]. Their main function is to extrude superfluous or harmful substances, such as metabolites, antibiotics, toxins, or metal ions. Some RNDs are also believed to be involved in export of siderophores and quorum sensing molecules [3,4], and there may be additional functions to be discovered, given the fact that the vast majority of RND pumps have not been characterized [5]. In general, the RNDs are divided into two groups depending on the substrates they transport: hydrophobic and amphiphilic efflux (HAE) and heavy metal efflux (HME).

**Competing interests:** The authors have declared that no competing interests exist.

Virtually all bacteria contain multiple RND assemblies with often at least partially overlapping functions. For instance, *Escherichia coli* contains six RNDs in its genome (five HAEs, transporting a broad range of substrates, and one HME, transporting Cu and Ag) [6], while the opportunistic pathogen *Pseudomonas aeruginosa* can contain up to 13 different RND systems, depending on the strain [7]. This abundance remains a puzzle. While in *E. coli* deletion of all RNDs results in drastic changes in the phenotype and seriously decreased ability to grow, deletion of one or two RND systems does not seem to have a strong effect (with the exception of the HME Cu-transporting Cus system, which is required for Cu-resistance) [6]. These results suggest functional overlap between the RND systems, and the pumps may be expressed depending on circumstances like exponential/stationary phase or aerobic/anaerobic conditions.

Classically, the efflux system is formed as a tripartite assembly [8]. Most RND systems share the same architecture, with an RND homotrimer in the inner membrane bound to six protomers of a membrane fusion protein (MFP) in the periplasm, which in turn connect the assembly with a trimer of outer membrane proteins (OMP). However, there are exceptions. In the MdtABC (multidrug transport) system from *E. coli* the inner membrane part is formed by a heterotrimer of $MdtB_2C_1$ stoichiometry [9]. A similar system called MuxABC was described in *P. aeruginosa*, with the MuxA, MuxB and MuxC proteins being homologous to MdtA, MdtB and MdtC, respectively (40, 65 and 61% sequence identity, and 78, 91 and 88% sequence similarity between the corresponding proteins) [10,11]. Other homologous systems were found and characterized in *Salmonella enterica* Serovar Typhimurium [12], *Serratia marcescens* [13], *Erwinia amylovora* [14], *Pseudomonas putida* [15], and *Photorhabdus luminescens* [16].

There are conflicting reports in the literature in regard to functional flexibility between the two subunits. Kim et al. reported that deletion of MdtC, but not MdtB, completely abolished the function of the Mdt system [9], while Da Wang and Fierke showed the opposite to be true [17]. It is possible there is a partial functional overlap between the two proteins, yet both subunits are needed for full function. Interestingly, the Mdt system has been shown to be able to facilitate both heavy metal and hydrophobic and amphiphilic efflux, and the heterogeneity of the inner membrane components may be a source of this promiscuity [17].

The evolutionary history of the two-RND subunit systems such as MdtBC remains unknown. While it may be hypothesized they arose originally through a gene duplication of the progenitor Mdt gene, both due to their high sequence similarity (e.g. 50% between MdtB and MdtC, compared to 25–30% between MdtB and other RNDs in *E. coli)* and their adjacent positions in genomes, it is not known if this gene duplication happens commonly in bacterial genomes or if it is rather an older phenomenon. The two-RND subunit systems from *E. coli* and *P. aeruginosa* are quite similar to each other, with higher homology between MdtB and MuxB, and between MdtC and MuxC, than between the proteins from the same organisms. *This* observation suggests that the original RND gene duplication might indeed be an infrequent older phenomenon, and not a widespread feature happening frequently in bacterial genomes.

Within the highly diverse Proteobacteria, Epsilonproteobacteria separated earliest from the rest, in an event placed at around 2.8 bln years ago by Battistuzzi and Hedges [25]. Subsequently, Deltaproteobacteria split from the rest of the lineage around 2.6 bln years ago, and Alphaproteobacteria around 2.4 bln years ago. The split between the two last groups, Beta- and Gammaproteobacteria, happened the latest, around 1.6 bln years ago. We set out to investigate how the phylogenetics of the Mdt proteins compares to the evolution of the phylum, in order to shed light on the evolutionary history of the Mdt systems. We thus performed a number of phylogenetic analyses and present the results in this paper.

## Materials and methods

### Phylogenetic analyses

In order to place the Mdt proteins in the context of other RNDs, the RND sequences from the work of Godoy et al. were used [5]. Out of over 2000 sequences there, 1106 were identified in UniProt (a full list is provided in the S1 File). These sequences were aligned with MAFFT using the default settings [18]. The alignment was then used to construct a phylogenetic tree based on all non-gapped positions and using neighborhood joining. The heterogeneity among sites was estimated by the MAFFT algorithm and the bootstrap values were calculated from 100 replicates.

The sequences that clustered together with *E. coli* MdtB and MdtC (and *E. coli* AcrB as an outgroup) were aligned with MAFFT using G-INS-i, an iterative refinement method, and a phylogenetic tree was constructed using neighborhood joining (NJ) of all of gap-free sites (JTT substitution model, the heterogeneity among sites was estimated by MAFFT, and bootstrap of 100 was used) [18]. The tree was then rooted on AcrB.

### Sequence similarity network and genomic neighborhood diagrams

The sequence similarity network (SSN) was generated with the Enzyme Similarity Tool (ESI-EST) and visualized with Cytoscape [20–24], with an alignment score of 200. The genomic neighborhoods of the genes in Fig 2 were investigated with the Gene Neighborhood Tool (ESI-GNT) [20–23], and visualized together with the phylogenetic trees in iTOL [19].

The sequences for membrane fusion proteins and outer membrane proteins were identified with the help of the ESI-GNT, and the further analysis was done in the same way as for RND proteins, using *E. coli* AcrA and TolC as outgroups, respectively. The phylogenetic trees were visualized with iTOL.

## Results & discussion

### RNDs form a number of distinct clusters

The comparison of over 1000 sequences of RND proteins, previously identified by Godoy et al. [5], was performed in order to divide them into functional groups, and thus clarify their possible evolutionary origins. In particular, we were interested in how the Mdt system is placed in relation to the better characterized efflux pumps like Acr, Mex (HAE) or Cus (HME). Since constructing reliable sequence alignments of large proteins containing both transmembrane helices and large periplasmic domains can be difficult, we also generated a sequence similarity network (SSN), to visualize direct relationships between the sequences [20–24].

There was a high similarity between the results obtained with the traditional phylogenetic analysis and the SSN. As seen in Fig 1, most proteins formed several large branches and clusters, with a smaller number remaining separated. The largest cluster (cluster 1) encompassed most of the characterized RNDs (all HAEs from *E. coli*). The less studied RNDs from *P. aeruginosa* clustered as MexI/W (cluster 2) and TriC/MexK (cluster 5). As expected, the HME proteins clustered together, with a further subdivision into mono- and di- valent transporting RNDs (cluster 3).

The Mdt proteins formed a distinct cluster (cluster 4), with one of the longest branches from the middle in the phylogenetic tree. The MdtB-like and MdtC-like proteins split early in the phylogenetic tree. The MdtB-like proteins, which are always directly adjacent to their respective MFPs, clustered together into one branch. The MdtC-like protein, which are never directly adjacent to their respective MFPs (i.e. there is always an MdtB-like protein in between), also clustered together into one branch. The fact that the gene organization has been

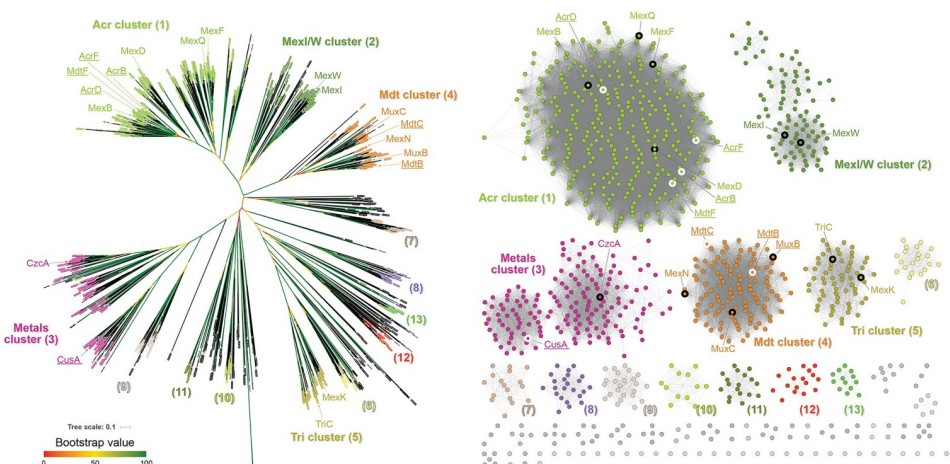

**Fig 1. Analysis of >1000 RND sequences.** Left, an unrooted phylogenetic tree, with bootstrap values represented as branch colors. Leaves are labelled with colors corresponding to their clusters (black represents proteins not belonging to the colored clusters, or singletons). Right, sequence similarity network. Clusters containing more than 10 sequences are colored and given a number. *P. aeruginosa* and *E. coli* proteins are marked with thick circles, black and white, respectively. On both panels *P. aeruginosa* and *E. coli* proteins are additionally marked with their abbreviations (*E. coli* proteins underlined).

preserved corroborates the notion that this heteromeric RND system is a result of an ancient gene duplication. This relationship seems to be very old, since Alpha-, Beta- and Gammaproteobacterial MdtBs and MdtCs form separate clusters, so that would put this duplication event to be older than the split between the major groups of *Proteobacteria* (over 2 billion years ago [25]).

Surprisingly, the branch/cluster containing Mdt-like proteins also included other RNDs, notably the MexN from *P. aeruginosa* and its homologues from other *Pseudomonodales*, as well as a number of other proteins. To investigate if this was an artefact caused by aligning a large number of sequences, we performed a new multiple sequence alignment with these 126 sequences, with *E. coli* AcrB as an outgroup. The results are shown in Fig 2, together with their genomic neighborhoods.

## A closer look into the Mdt cluster reveals the evolutionary history of the subfamily

As observed in the analysis of all RNDs (Fig 1), the MexN-like proteins clustered together with MdtB- and MdtC-like proteins (Fig 2). However, an interesting observation was that the MexN-like proteins were divided into two distinct groups. The first group was formed by MexN-like proteins from strains that also contained an MdtBC system. These MexN-like proteins clustered together with MdtB-like proteins, suggesting their common evolutionary origin (i.e. these MexN-like proteins and MdtB-like proteins are descendant from one of the originally duplicated genes). The second group was formed by MexN-like proteins from strains that did not contain an MdtBC system, and these MexN-like proteins separated from the rest of the tree before the split between MdtB- and MdtC-like proteins. Because this second group of RNDs split earliest from the rest, it is likely that they are directly descendant from the progenitor single RND, and no gene duplication occurred during their evolution. Since this subset of MexN-like proteins never underwent the gene duplication event, we subsequently named them "progenitor-like" RNDs in order to distinguish them from the "true" MexN-like

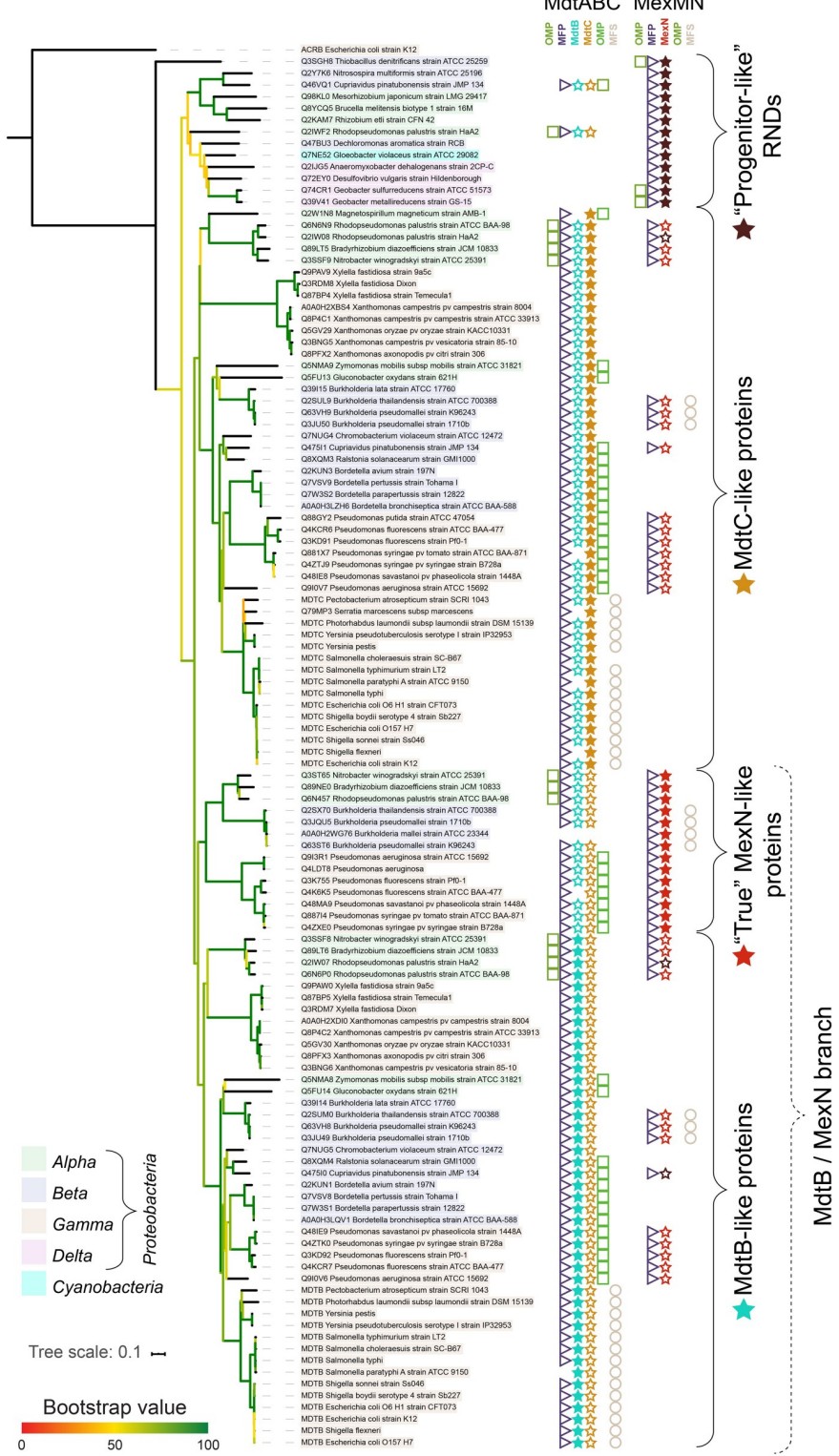

**Fig 2. A phylogenetic tree of the Mdt-like proteins.** The bootstrap values are represented by a branch color as in Fig 1 (the branches with bootstrap support lower than 50 were not collapsed, in order to show the genomic neighborhood of these genes). The taxonomy of each organism is represented with shading of the labels. To the right of the protein and organism names the genomic context is presented. The actual protein at the leaf tip is represented with a filled symbol: a dark red star for "progenitor-like" RNDs, a red star for "true" MexN-like proteins, a blue star for MdtB-like

proteins and an orange star for MdtC-like proteins). The open symbols provide the genomic context for the RNDs. For example, in the second row, *Thiobacillus denitrificans*, the lack of symbols under "MdtABC" means there are no proteins from this group present in this organism. Further to the right under "MexMN", an open green square means there is an OMP present, followed by an MFP (an open purple triangle), and an RND (a closed star). Lack of symbols under "OMP" and "MFS" means there are no further proteins in this set of genes.

proteins, with "true" meaning here "clustering together with *P. aeruginosa* MexN and therefore having the same evolutionary history".

The "progenitor-like" RND group contained all Deltaproteobacterial sequences represented in our analysis, as well as the only sequence from a non-Proteobacterium, *Gloeobacter violaceus*, a Cyanobacterium (Fig 2). The fact that these "progenitor-like" RNDs did not cluster together with known MdtB- and MdtC-like proteins suggests they are direct descendants of the ancient common ancestor of the whole Mdt cluster, the progenitor gene. We also performed searches for MdtB- and MdtC-like proteins (i.e. having sequence similarity at least 40%) in Deltaproteobacteria and found only three hits, suggesting the MdtBC-like systems are virtually absent in these two groups. A number of Alpha- and Betaproteobacterial orders contained the "progenitor-like" RNDs, but no Gammaproteobacteria did. Interestingly, all the Alpha- and Betaproteobacterial representants can fix nitrogen and/or reduce nitrate, suggesting a common habitat [26]. The fact that all the older bacterial lineages appeared in this group suggests that the original gene duplication that produced MdtB- and MdtC-like proteins occurred in the common ancestor of the Alpha-, Beta- and Gammaproteobacteria, around the end of the Archean Eon [25], and the sporadic occurrence of a "progenitor-like" RND in Alpha- and Betaproteobacteria is more likely a result of a horizontal gene transfer.

The rest of the RNDs formed two groups, with all the MdtB-like proteins in one and all the MdtC-like proteins in the other. Noticeably, the "true" MexN-like proteins clustered together with the MdtB-like proteins. This observation suggested the MexN separation happened after the original gene duplication that formed MdtB and MdtC from the progenitor RND gene. In general, the branching of both MdtB and MdtC groups was similar: Alphaproteobacteria separated earliest (with the exception of *Gluconobacter oxydans* and *Zymomonas mobilis*, see below), and then Beta- and Gammaproteobacteria. Surprisingly, the Gammaproteobacterial order *Xanthomonodales* separated together with Alphaproteobacteria (both in the MdtB- and the MdtC-like groups, with moderate to low bootstrap support, however). In Alphaproteobacteria, homologs of MdtBC/MexN were numerously found only in orders *Rhizobiales* and *Rhodospirillales*, and sporadically in a few other orders. In Betaproteobacteria, homologs of MdtBC/MexN were widespread and found in all major orders, and in Gammaproteobacteria homologs of MdtBC/MexN were found in most orders. In all three major Proteobacterial families there were examples of closely related species and strains where one contained MdtBC, MexN or both, and the other with no MdtBC/MexN homologs. In many organisms it was also suspected the process of losing the RND pumps was ongoing. For instance, in *Shigella flexneri*, *Serratia marcescens*, *Pseudomonas syringae pv tomato* and *Magnetospirillum magneticum* an MdtB was missing; in *Salmonella paratyphi A* an MFP was missing; and in *Burkholderia mallei* the whole MdtABC operon was absent (see S1 File for details).

The genomic neighborhoods provided additional insights into the evolutionary history of the Mdt systems. Among the MdtBC systems, many contained OMP components, and the architecture was conserved in the main groups: in Alphaproteobacteria the OMP preceded the MFP, and in Beta- and Gammaproteobacteria it followed the MdtC protein. It is likely that the OMP components were acquired after the original gene duplication and this acquisition happened separately, once in Alphaproteobacteria, and once in a common ancestor to Beta- and Gammaproteobacteria, and in many cases it was subsequently lost (see S1 File for details).

Moreover, all *Enterobacterales* possessed an additional inner membrane protein from the Major Facilitator Superfamily (MFS), called MdtD in *E. coli*, an iron and citrate exporter [27], and no outer membrane proteins. The order *Enterobacterales* is an example of how the outer membrane channel function had converged on just one protein (e.g. TolC in *E. coli*), and the redundant outer membrane components of RND systems are removed from the genomes (with the exception of specialized functions, e.g. *E. coli* CusC as an outer membrane component for the Cu-exporting Cus system). The outer membrane proteins were also missing in the order *Xanthomonodales* and sporadically in other organisms. Notably, the *Burkholderia* MexMNs also contained an MFS, not related to other MFSs observed here.

## Horizontal gene transfers

The exception to the observation that organisms containing a "progenitor-like" RND did not contain an MdtBC system occurred in *Cupriavidus pinatubonensis* (*Betaproteobacteria*, order *Burkholderiales*), and one of two strains of *Rhodopseudomonas palustris*, namely strain HaA2 (*Alphaproteobacteria*, order *Rhizobiales*). These two organisms possessed both an MdtBC-like system, similar to other Proteobacteria in their respective groups, and a "progenitor-like" RND, likely a result of a horizontal gene transfer. The *C. pinatubonensis* RND showed close similarity to an RND from *Nitrosospira multiformis*, a distantly related Betaproteobacterium (order *Nitrosomonadales*), and their "progenitor-like" RNDs grouped together with other "progenitor-old" RNDs. The *R. palustris* HaA2 strain possibly lost the original MexN-like system and incorporated a "progenitor-like" RND, judging from its genomic contexts (see S1 File). The other *R. palustris* strain, ATCC BAA-871, did not contain a "progenitor-like" RND system, and its other MdtBC- and MexN-like proteins behaved as its relatives in other Alphaproteobacteria.

A number of sequences originally clustering with other Mdts in Fig 1 did not align well and in consequence showed poor or unresolved phylogeny with low bootstrap values regardless of the methods used and were therefore removed from the analysis prior to the results shown in Fig 2. These sequences are described in the S1 File.

Proteins from two Alphaproteobacteria, *Gluconobacter oxydans* and *Zymomonas mobilis*, did not cluster together with other Alphaproteobacterial Mdts, but were found closest to respective proteins from the order *Burkholderia*. While the long branches observed for all four proteins as well as moderate bootstrap values might render this clustering less reliable, it is possible those two organisms had lost their original Mdts and acquired new ones via horizontal gene transfer. Moreover, *G. oxydans* possesses a third protein with high sequence similarity to its own MdtC (not shown in Fig 2, see S1 File). It is likely a result of a discrete gene duplication, particularly since this third gene does not possess an MFP.

As mentioned above, the order *Xanthomonadales* clustered somewhat reliably with Alphaproteobacteria, both in MdtB- and MdtC-like groups. They did not possess a third RND, either a "true" MexN-like protein or a "progenitor-like" RND. Since they separated the earliest from other Gammaproteobacteria, it is possible their ancestors lost both their original MdtABC and MexMN systems, and subsequently incorporated an MdtABC from an Alphaproteobacterium [28].

## Reconstructing the MdtABC/MexMN evolution

The results described here, together with analysis of corresponding MFPs (see S1 File) made it possible to propose an evolutionary scenario for the appearance of MdtBC and MexN pumps (Fig 3). The original RND progenitor gene underwent a duplication in the common ancestor to Alpha-, Beta- and Gammaproteobacteria, while remaining single in other bacterial groups

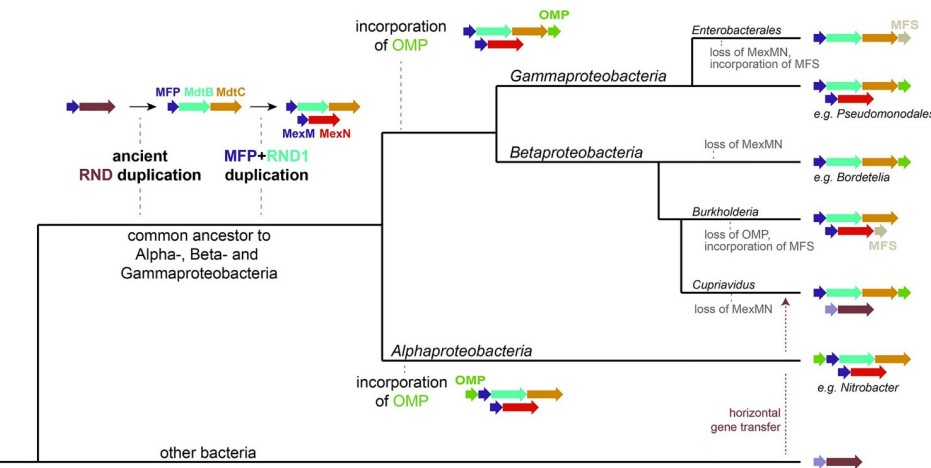

**Fig 3. The proposed evolutionary scenario.** The evolution within the Alpha-, Beta- and Gamma- proteobacteria groups is shown, as deduced from the phylogenetic tree in Fig 2 and a timeline of evolution of Proteobacteria (Battistuzzi, Feijao and Hedges 2004). The cladograms lengths and timepoints of evolutionary events are not to scale. As an example of horizontal gene transfer, *Cupriavidus pinatubonensis* is also shown.

(as "progenitor-like" RNDs). The MFP and the adjacent RND were duplicated, forming the "true" MexMN system. In the next step an OMP was acquired, and was inserted before the MFP in Alphaproteobacteria, or after the MdtC in the common ancestor to the Beta- and Gammaproteobacteria. From these points many organisms lost the MexMN system. In Alpha-proteobacteria only two orders represented in Figs 1 and 2 retained the original genes. Many Betaproteobacteria retained the OMPs (occurring always after the MdtCs) but lost the dupli-cated MexMN, with the exception of the *Burkholderia* genus, which lost the OMPs, but retained the MexMN and also gained an MFS next to it. In Gammaproteobacteria the configu-ration was generally kept intact, with the exception of *Enterobacterales*, which lost the MexMN and incorporated an MFS into its MdtABC operon.

## Supporting information

**S1 File. Supporting information containing the list of used sequences, as well as detailed discussion, is available.**
(DOCX)

## Author Contributions

**Conceptualization:** Kamil Górecki.

**Formal analysis:** Kamil Górecki.

**Funding acquisition:** Megan M. McEvoy.

**Investigation:** Kamil Górecki.

**Methodology:** Kamil Górecki.

**Project administration:** Megan M. McEvoy.

**Supervision:** Megan M. McEvoy.

**Validation:** Megan M. McEvoy.

**Writing – original draft:** Kamil Górecki.

**Writing – review & editing:** Kamil Górecki, Megan M. McEvoy.

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
