## [Decision Letter · Decision Letter 0]

13 Jan 2020

PONE-D-19-34793

Phylogenetic analysis reveals an ancient gene duplication as the origin of the MdtABC efflux pump

PLOS ONE

Dear Dr. Gorecki,

Thank you for submitting your manuscript to PLOS ONE. After careful consideration, we feel that it has merit but does not fully meet PLOS ONE’s publication criteria as it currently stands. Therefore, we invite you to submit a revised version of the manuscript that addresses the points raised during the review process.

As you see from the two critiques, both reviewers were highly enthusiastic about your paper, which I share.  Their concerns are minor and editorial in nature.

We would appreciate receiving your revised manuscript by February 13th, 2020. To enhance the reproducibility of your results, we recommend that if applicable you deposit your laboratory protocols in protocols.io, where a protocol can be assigned its own identifier (DOI) such that it can be cited independently in the future. For instructions see: http://journals.plos.org/plosone/s/submission-guidelines#loc-laboratory-protocols

We look forward to receiving your revised manuscript.

Kind regards,

William M Shafer, Ph.D.

Academic Editor

PLOS ONE

Journal Requirements:

Reviewers' comments:

Reviewer's Responses to Questions

**Comments to the Author**

1. Is the manuscript technically sound, and do the data support the conclusions?

Reviewer #1: Yes

Reviewer #2: Partly

2. Has the statistical analysis been performed appropriately and rigorously? 

Reviewer #1: Yes

Reviewer #2: N/A

3. Have the authors made all data underlying the findings in their manuscript fully available?

Reviewer #1: Yes

Reviewer #2: Yes

4. Is the manuscript presented in an intelligible fashion and written in standard English?

Reviewer #1: Yes

Reviewer #2: Yes

5. Review Comments to the Author

Reviewer #1: This manuscript by Gorecki and McEvoy examines the evolutionary history of RND efflux pumps. The authors examine the Mdt efflux system which uniquely consists fo two RND subunits (MdtBC). Through genomic analyses, the work makes a solid case that this two protein arrangement arose through an ancient gene duplication event. Moreover, the authors reveal an evolutionary relationship between MdtBC and the MexN RND subunit. The authors reconstruct a likely history of how these RND subunits evolved via gene duplication and subsequent acquisition of additional membrane fusion proteins and outer membrane proteins. A pleasing outcome of these analyses is the observation that loss of MdtB occurs in preference to loss of MdtC; so, it is likely that MdtC can at least partially perform MdtB function while MtdB may not have the ability to function in stead of MdtC. This findings will inform current thinking on the relative importance of MdtBC subunits.

I only have minor text changes to suggest:

-The difference between “old” and “true” MexN could be better explained. Symbols in Fig 2 also need better explanation - what exactly is the difference between filled and opened symbols?

-Line 237 “Fig. s 1” typo

-Line 238: “post OMPs” is not a very clear description of (I assume) genomic organization.

Reviewer #2: This paper presents an interesting hypothesis for the origin of the two component resistance nodulation division pump, MdtABC. The work is based upon a freely available dataset published in 2010 containing over 2000 RND gene sequences, providing good representation of RND gene sequences found across populations. In general, more explanation is needed throughout the manuscript to clarify the meaning of the text and justify the conclusions drawn. With additional explanation in certain parts, we feel this paper will add to our understanding of the origin and conservation of two component RND pumps, across Gram-negative bacteria.

The grammar and punctuation should be reviewed throughout (e.g. line 144).

The authors state with confidence that MdtABC arose from a gene duplication. In line 65, the authors state that it is “obvious” that MdtBC arose through gene duplication. While we agree this is highly likely, we aren’t sure it is fair to say it obvious without providing any justification or evidence for this. This section should be reworded and justification for this assertion added.

Line 70 The authors refer to the gene duplication event being a “rare older phenomenon”. While we think we understand what the authors are implying we feel this suggests that such gene duplication events would/can no longer occur. Perhaps it would be better to say it is an infrequent occurrence.

The methods section is rather brief and more information is required throughout to improve clarity. For example on line 80, is the algorithm estimated by MAFFT or are the authors referring to something else? Should a reference be added for the Gene Neighbourhood tool? Also in the methods section, the mex genes are mentioned but have not previously been introduced. These should be added to the introduction.

Line 122 – “Indeed, all proteins adjacent to their respective MFPs cluster into one branch, and all the other ones form another branch.” It isn’t clear what it meant by this statement and more detail/explanation is needed to understand the conclusions being drawn from the phylogenetic tree.

The use of “old” to describe ancestral genes is confusing, and slightly misleading. The authors state the origin of a subset of the MexN genes is “likely” older than the gene duplication event so they named them “old”RNDs. We think these should be renamed and that further justification for these assumptions given. We don’t necessarily disagree but it is important that the conclusions drawn from the data are clear and justified.

It is not clear what a ‘typical MdtABC’ pump means? If a particular consensus sequence has been used then it should be stated or referenced.

To help readers understand the relevance of the gene variation between proteobacteria strains, perhaps it would be useful to add some information in the introduction about the evolution of the phylum.

Minor comments

Inconsistent abbreviations are used for proteins mentioned throughout (in lines 13, 16 52, 81, 143, 167, 168). MdtB is frequently referred as so, however, MdtC commonly just referred to as “C” but would be clearer if spelt out as MdtC.

MdtB and MdtC referred to as “Mdt pumps”, yet MexMN that form the same sort of complex referred to as one unit (line 19). Perhaps this could be edited to provide consistency.

Abbreviation of “sequence similarity network” is not consistent. First introduced as SSN on line 87, then described by full name on 90, then mentioned again as “sequence similarity network (SSN)” on 101.

Typo on line 188 - Beta+Gammaproteobacteria?

Both figures 1 and 2 are very blurry the text in figure 2 is very small when printed.

Sentence from 143 to 147 could be clarified. I think if the sentence is split into two, it would be easier to follow and understand the differences between MdtABC positive and negative strains.

On line 152, are the authors referring to figure 2? The text is slightly ambiguous and could be clarified.

6. PLOS authors have the option to publish the peer review history of their article (what does this mean?). If published, this will include your full peer review and any attached files.

Reviewer #1: No

Reviewer #2: Yes: Jessica Blair, Hannah Pugh

---

## [Author Response · Author response to Decision Letter 0]

22 Jan 2020

(as in the Response to reviewers document)

We thank the reviewers for their careful consideration and suggestions to improve the manuscript. We’ve incorporated their suggested changes as described below. We feel that with these changes the manuscript is much improved and hope that it is now considered suitable for publication in PLoS ONE.

Reviewer #1: This manuscript by Gorecki and McEvoy examines the evolutionary history of RND efflux pumps. The authors examine the Mdt efflux system which uniquely consists fo two RND subunits (MdtBC). Through genomic analyses, the work makes a solid case that this two protein arrangement arose through an ancient gene duplication event. Moreover, the authors reveal an evolutionary relationship between MdtBC and the MexN RND subunit. The authors reconstruct a likely history of how these RND subunits evolved via gene duplication and subsequent acquisition of additional membrane fusion proteins and outer membrane proteins. A pleasing outcome of these analyses is the observation that loss of MdtB occurs in preference to loss of MdtC; so, it is likely that MdtC can at least partially perform MdtB function while MtdB may not have the ability to function in stead of MdtC. This findings will inform current thinking on the relative importance of MdtBC subunits.

I only have minor text changes to suggest:

-The difference between “old” and “true” MexN could be better explained. 

In accordance to similar comments from Reviewer 2, we modified the text to bring more clarity to the discussed difference and stress the difference between the two sets of MexN-like proteins (see below). 

- Symbols in Fig 2 also need better explanation - what exactly is the difference between filled and opened symbols?

We expanded the figure legend to provide more detailed description of the symbols, and also included an example of how to read the figure. 

-Line 237 “Fig. s 1” typo

The typo in line 237 was corrected. 

-Line 238: “post OMPs” is not a very clear description of (I assume) genomic organization.

We corrected the text in line 238 to clearly reflect that we mean the genomic organization. 

Reviewer #2: This paper presents an interesting hypothesis for the origin of the two component resistance nodulation division pump, MdtABC. The work is based upon a freely available dataset published in 2010 containing over 2000 RND gene sequences, providing good representation of RND gene sequences found across populations. In general, more explanation is needed throughout the manuscript to clarify the meaning of the text and justify the conclusions drawn. With additional explanation in certain parts, we feel this paper will add to our understanding of the origin and conservation of two component RND pumps, across Gram-negative bacteria.

The grammar and punctuation should be reviewed throughout (e.g. line 144).

We reread the manuscript and corrected the mistakes. 

The authors state with confidence that MdtABC arose from a gene duplication. In line 65, the authors state that it is “obvious” that MdtBC arose through gene duplication. While we agree this is highly likely, we aren’t sure it is fair to say it obvious without providing any justification or evidence for this. This section should be reworded and justification for this assertion added.

We modified the section to reflect the still hypothetical nature of our notion at that point of the manuscript. We also slightly modified the paragraph regarding the results shown in Fig. 1: the fact that all MdtBs and MdtCs clustered together corroborates the notion that they arose from a single gene via gene duplication. 

Line 70 The authors refer to the gene duplication event being a “rare older phenomenon”. While we think we understand what the authors are implying we feel this suggests that such gene duplication events would/can no longer occur. Perhaps it would be better to say it is an infrequent occurrence.

We agree, and we changed the wording accordingly from “rare” to “infrequent” in line 70.

The methods section is rather brief and more information is required throughout to improve clarity. For example on line 80, is the algorithm estimated by MAFFT or are the authors referring to something else? Should a reference be added for the Gene Neighborhood tool? Also in the methods section, the mex genes are mentioned but have not previously been introduced. These should be added to the introduction.

We edited the methods section to provide more information: the algorithm was by MAFFT; The Gene Neighborhood Tool is part of the toolset provided by the authors of Enzyme Similarity Tool, we therefore repeated the references to make this clear. And we also clarified which genes neighborhoods were investigated by removing the “mex” acronym in the methods section. 

Line 122 – “Indeed, all proteins adjacent to their respective MFPs cluster into one branch, and all the other ones form another branch.” It isn’t clear what it meant by this statement and more detail/explanation is needed to understand the conclusions being drawn from the phylogenetic tree.

We modified this section to better explain the line of thinking. We described how the genomic organization (MdtB being directly adjacent to an MFP) was in agreement with sequence similarities and the conclusion drawn from that fact being the gene duplication happened once and the order of the genes was preserved. 

The use of “old” to describe ancestral genes is confusing, and slightly misleading. The authors state the origin of a subset of the MexN genes is “likely” older than the gene duplication event so they named them “old”RNDs. We think these should be renamed and that further justification for these assumptions given. We don’t necessarily disagree but it is important that the conclusions drawn from the data are clear and justified.

We agree with the Reviewer 2 that the word “old” can be misleading, and also took into consideration a similar comment from Reviewer 1. In this paper we aim to distinguish between single subunit RNDs that have a history of gene duplication (like P. aeruginosa MexN, through a gene duplication) and single RNDs that are directly descendant from the original progenitor Mdt gene (and not through any gene duplication). We decided to keep the word “true” for describing the RNDs that are like the P. aeruginosa MexN, in the meaning of having the same evolutionary history (i.e. first via the gene duplication of the progenitor RND into two component RND, and then subsequent MFP+RND1 gene duplication). In order to avoid using a common word “old”, we chose to name the second group “progenitor-like” RNDs, to underline their direct resemblance to the progenitor RND Mdt gene and lack of any gene duplication history. We modified the text throughout accordingly and included more detailed explanation both in the Introduction and Results sections. 

It is not clear what a ‘typical MdtABC’ pump means? If a particular consensus sequence has been used then it should be stated or referenced.

By ‘typical’ we meant a pump consisting of two RND subunits. We modified the text in line 154 to reflect that by stating the 40% sequence similarity cut-off. 

To help readers understand the relevance of the gene variation between proteobacteria strains, perhaps it would be useful to add some information in the introduction about the evolution of the phylum.

We expanded the last paragraph of the introduction according to the suggestion and described how the Proteobacteria are believed to have evolved. 

Minor comments

Inconsistent abbreviations are used for proteins mentioned throughout (in lines 13, 16 52, 81, 143, 167, 168). MdtB is frequently referred as so, however, MdtC commonly just referred to as “C” but would be clearer if spelt out as MdtC.

We spelled out MdtC throughout the text to avoid confusion.

MdtB and MdtC referred to as “Mdt pumps”, yet MexMN that form the same sort of complex referred to as one unit (line 19). Perhaps this could be edited to provide consistency.

We changed the wording to “MdtBC” pumps where we mean “a system containing two RND subunits”, in order to distinguish it from a possible misunderstanding of “Mdt pumps” as “MdtBC + MexN” systems. 

Abbreviation of “sequence similarity network” is not consistent. First introduced as SSN on line 87, then described by full name on 90, then mentioned again as “sequence similarity network (SSN)” on 101.

We corrected the inconsistency by changing the wording in lines 87, 90 and 101. 

Typo on line 188 - Beta+Gammaproteobacteria?

This is not a typo, we meant here a common ancestor to both Beta- and Gammaproteobacteria (therefore a plus sign). We modified the text to make it clear what we mean by spelling it out as “a common ancestor to both Beta- and Gammaproteobacteria” in line 188. 

Both figures 1 and 2 are very blurry the text in figure 2 is very small when printed.

We believe the low quality of the figures is due to PLOS One’s PDF rendering. Upon clickling on “Click here to download Figure X” in the top right corner of a respective figure page, a high-resolution figure can be downloaded. The text size in Fig. 2 can unfortunately not be increased without splitting the figure into two pages. We prefer to avoid that, since the main message of Fig 2. is to show the clustering of proteins of similar genomic organization and comparison to the taxonomy of the organisms. The figures were also uploaded to PACE and adjusted accordingly.

Sentence from 143 to 147 could be clarified. I think if the sentence is split into two, it would be easier to follow and understand the differences between MdtABC positive and negative strains.

We expanded this section to more clearly describe the difference between the various groups of proteins, also to meet the similar request of Reviewer 1. 

On line 152, are the authors referring to figure 2? The text is slightly ambiguous and could be clarified.

Yes, we are referring to Figure 2. We modified the text to improve clarity by adding a reference.

---

## [Editor Report · Decision Letter 1]

27 Jan 2020

Phylogenetic analysis reveals an ancient gene duplication as the origin of the MdtABC efflux pump

PONE-D-19-34793R1

Dear Dr. Gorecki,

We are pleased to inform you that your manuscript has been judged scientifically suitable for publication and will be formally accepted for publication once it complies with all outstanding technical requirements.

With kind regards,

William M Shafer, Ph.D.

Academic Editor

PLOS ONE
---

## [Editor Report · Acceptance letter]

31 Jan 2020

PONE-D-19-34793R1 

Phylogenetic analysis reveals an ancient gene duplication as the origin of the MdtABC efflux pump 

Dear Dr. Górecki:

I am pleased to inform you that your manuscript has been deemed suitable for publication in PLOS ONE. Congratulations! Your manuscript is now with our production department. 

With kind regards,

on behalf of

Professor William M Shafer 

Academic Editor

PLOS ONE